# Effect of Inflammatory Microenvironment on the Regenerative Capacity of Adipose-Derived Mesenchymal Stem Cells

**DOI:** 10.3390/cells12151966

**Published:** 2023-07-29

**Authors:** Diána Szűcs, Vanda Miklós, Tamás Monostori, Melinda Guba, Anikó Kun-Varga, Szilárd Póliska, Erika Kis, Balázs Bende, Lajos Kemény, Zoltán Veréb

**Affiliations:** 1Regenerative Medicine and Cellular Pharmacology Laboratory, Department of Dermatology and Allergology, University of Szeged, 6720 Szeged, Hungary; szucs.diana@med.u-szeged.hu (D.S.); monostori.tamas.bence@med.u-szeged.hu (T.M.); guba.melinda@med.u-szeged.hu (M.G.); kun-varga.aniko@med.u-szeged.hu (A.K.-V.); kemeny.lajos@med.u-szeged.hu (L.K.); 2Doctoral School of Clinical Medicine, University of Szeged, 6720 Szeged, Hungary; 3Centre of Excellence for Interdisciplinary Research, Development and Innovation, University of Szeged, 6720 Szeged, Hungary; 4Biobank, University of Szeged, 6720 Szeged, Hungary; miklos.vanda@szte.hu; 5Genomic Medicine and Bioinformatics Core Facility, Department of Biochemistry and Molecular Biology, Faculty of Medicine, University of Debrecen, H-4032 Debrecen, Hungary; poliska@med.unideb.hu; 6Dermatosurgery and Plastic Surgery, Department of Dermatology and Allergology, University of Szeged, 6720 Szeged, Hungary; kis.erika.gabriella@med.u-szeged.hu (E.K.); bende.balazs@med.u-szeged.hu (B.B.); 7Hungarian Centre of Excellence for Molecular Medicine-USz Skin Research Group, University of Szeged, 6720 Szeged, Hungary

**Keywords:** adipose-derived mesenchymal stem cells, lipopolysaccharide, tumor necrosis factor α, inflammation, regenerative medicine

## Abstract

Adipose-derived mesenchymal stem cells are increasingly being used in regenerative medicine as cell therapy targets, including in the treatment of burns and ulcers. The regenerative potential of AD-MSCs and some of their immunological properties are known from in vitro studies; however, in clinical applications, cells are used in non-ideal conditions and can behave differently in inflammatory environments, affecting the efficacy and outcome of therapy. Our aim was to investigate and map the pathways that the inflammatory microenvironment can induce in these cells. High-throughput gene expression assays were performed on AD-MSCs activated with LPS and TNFα. Analysis of RNA-Seq data showed that control, LPS-treated and TNFα-treated samples exhibited distinct gene expression patterns. LPS treatment increased the expression of 926 genes and decreased the expression of 770 genes involved in cell division, DNA repair, the cell cycle, and several metabolic processes. TNFα treatment increased the expression of 174 genes and decreased the expression of 383 genes, which are related to cell division, the immune response, cell proliferation, and differentiation. We also map the biological pathways by further investigating the most altered genes using the Gene Ontology and KEGG databases. Secreted cytokines, which are important in the immunological response, were also examined at the protein level, and a functional assay was performed to assess wound healing. Activated AD-MSC increased the secretion of IL-6, IL-8 and CXCL-10, and also the closure of wounds. AD-MSCs presented accelerated wound healing under inflammation conditions, suggesting that we could use this cell in clinical application.

## 1. Introduction

Adipose tissue can be found in several locations in the human body, such as subcutaneous and visceral sites, intra-articular, intramuscular, intra-hepatic depots, and in bone marrow. Adipose tissue is not only an energy reservoir but also an endocrine organ since it produces mediators of metabolism and cell function. Adipocytes and nonadipocytes secrete adipokines (leptin, adiponectin, omentin, and resistin), pro- and anti-inflammatory cytokines (IL-6, TNF-a, IL-1b, IL-8, MCP-1, IL-1Ra, IL-6, IL-7, IL-8 and IL-11), growth factors (VEGF, HGF, FGF, IGF-1 and BDNF), pro-apoptotic factors, pro-angiogenic factors and microvesicles filled with proteins and nucleic acids. Three types of adipose tissue can be distinguished: (I) white adipose tissue, which has a role in energy storage but also produces adipokines; (II) brown adipose tissue, which plays a role in thermogenesis regulation but can also store energy; and (III) beige adipose tissue, which is also included in thermogenesis and energy storage [1,2,3,4].

Adipose-tissue-derived mesenchymal stem cells (AD-MSCs) are located in adipose tissue, mainly in the stromal vascular fraction (SVF), which can be isolated via less invasive methods. AD-MSCs are multipotent cells with self-renewal capacity, and they have a multilineage ability to differentiate into cells of mesodermal lineages, such as adipocytes, chondrocytes, and osteoblasts. They have high proliferation and immunosuppressive properties; thus, they and even their secretome can be applied in regenerative medicine in diseases related to immune diseases. These cells have the potential to regulate the immune response since they can interact with several immune cells through direct cell–cell interactions or indirectly with their secretome. AD-MSCs have the ability to interact with T cells, B cells, macrophages, natural killer cells (NKs), dendritic cells (DCs), neutrophils, and mast cells [1,2,3,4,5,6,7,8,9].

AD-MSCs have an immunomodulatory effect through T-cell interactions via cell adhesion molecules and change the level of secreted mediators (IDO, TGFβ, IL-10, and PGE2). Furthermore, T cells can influence AD-MSCs by producing chemokines. Several studies showed, in the presence of a high amount of pro-inflammatory cytokines, that AD-MSCs activate Treg generation and inhibit T cell proliferation, activation, and differentiation, thus suppressing the immune response. On the other hand, in the presence of a low amount of pro-inflammatory cytokines, AD-MSCs inhibit Treg generation and activate T-cell proliferation, activation, and differentiation. AD-MSCs have an effect on B cells by suppressing and promoting their proliferation, activation, and differentiation and activating chemotaxis and Breg induction. They can inhibit NK cell proliferation, activation, and migration and induce NK cell progenitor proliferation and NK activation. Furthermore, AD-MSCs suppress DC differentiation, endocytosis, maturation, activation, and migration and inhibit mast cell degranulation, inflammatory cytokine expression, and chemotaxis. AD-MSCs can mediate macrophage polarization by suppressing the M1 and promoting the M2 phenotype, and they can influence neutrophils as well by inhibiting the activation, recruitment, formation of extracellular neutrophil traps, and protease secretion and promoting survival and recruitment [1,2,3,4,5,6,7,8,9,10,11,12].

Since AD-MSCs have immunomodulatory effects and angiogenic and differentiation potential, they provide the opportunity to replace, repair, and regenerate damaged tissues; thus, they can be applied in support of many diseases associated with tissue damage. AD-MSCs can be used on a wide scale for many different purposes, such as wound healing and skin regeneration (diabetic and non-diabetic ulcers and nonhealing wounds, extensive burns, and physicochemical skin injuries), autoimmune disorders, hematological disorders and graft-versus-host disease, bone and cartilage repair, cardiovascular and muscular diseases, neurodegenerative diseases, and radiation injuries. These cells provide an opportunity to recover the function of damaged tissues with high efficacy and safety. For therapeutic uses of AD-MSCs, their purity and potency must be identified prior to administration to ensure safe and successful application free from adverse events. The focus of our study is the wound healing and skin regeneration ability of AD-MSCs. In these diseases, AD-MSCs must regenerate tissues in a highly inflamed environment, which affects their expression profile. These experiments reflect the changes in gene and protein expression of AD-MSCs after exposure to inflammation agents [1,2,3,4,5,6,7,8,9,10,11,12,13,14,15,16,17,18,19,20].

The clinical treatment of chronic wounds and ulcers is a great challenge since the cells injected into the patient should regenerate the tissue in a non-physiological environment. Non-healing wounds are highly inflamed and often associated with bacterial infection, especially in diabetic patients. Thus, stem cells must survive and proliferate in an inflammatory microenvironment [6,7,13,21,22,23,24,25,26]. However, as stem cell therapies have become increasingly popular in recent years, the results of new therapies can be contradictory, and the injected cells are not pretreated, which can reduce the effectiveness. According to the literature, pretreatment of stem cells can increase their regenerative capacity and promote faster wound closure, so we believe that pretreatment of stem cells may be the key to developing a more effective therapy [16,27,28,29,30,31,32,33,34,35]. The aim of our study is to investigate the response of mesenchymal stem cells from adipose tissue to the inflammatory factors TNFα and LPS at the molecular and cellular levels. Our results may contribute to the development of a new therapy in which cell licensing plays a decisive role in increasing efficiency. Our experimental arrangement may also be suitable for examining the response of patient cells to inflammatory factors. In this way, we were able to test whether the licensing given is suitable for the given patient, thus helping to develop personalized therapy.

## 2. Materials and Methods

### 2.1. Isolation of SVF Fraction

The collection of adipose tissue was in accordance with the guidelines of the Declaration of Helsinki and was approved by the National Public Health and Medical Officer Service (NPHMOS) and the National Medical Research Council (16821-6/2017/EÜIG, STEM-01/2017), which follows the Directive 2004/23/CE of the EU Member States on the practice of presumed written consent for tissue collection. Abdominal adipose tissues were removed from the patients (Sex:2/3 F/M, Age: 50.2 ± 11.7 years), and the isolation was performed within 1 h after plastic surgery. Adipose tissue was homogenized and washed with Ca^2+^ and Mg^2+^ free PBS (Biosera, Nuaille, France), and then the sample was centrifuged at 600 rpm for 8 min at RT. The SVF fraction was located in the upper fraction, and it contains mesenchymal stem cells. After two steps of washing, tissue digestion was performed with (0.25 mg/mL) Collagenase Type IA (Merck KGaA, Darmstadt, Germany) for 1 h at 37 °C on a tube rotator. The collected cells were washed, and then the upper layer was removed to preserve only the SVF fraction at the bottom of the tube. The cell pellet was suspended in Ca^2+^ and Mg^2+^ free PBS (Biosera, Nuaille, France), and it was filtered using a 100 µm Corning^®^ cell strainer. After the washing step was repeated, the yellowish upper layer was kept and suspended in 1 ml of medium. Cells were counted using the EVE automatic cell counter, NanoEntek (NanoEntek, Seoul, Korea), then they were seeded in a T25 cm^2^ flask. For maintenance, DMEM-HG medium (Biosera, Nuaille, France) was applied supplemented with 10% FBS (Biosera, Nuaille, France), 1% L-glutamine (Biosera, Nuaille, France) and 1% Antibiotic–Antimycotic Solution (Biosera, Nuaille, France) for further experiments.

### 2.2. Differentiation of Adipose-Tissue-Derived Mesenchymal Stem Cells

The differentiation capacity of adipose-tissue-derived mesenchymal stem cells was validated by differentiation into adipocyte, chondrocyte and osteocyte lines. They were seeded in a 24-well plate, 5 × 10^4^ cells/well, and after 24 h, the medium was replaced with a differentiation medium. For this purpose, commercially available Gibco’s StemPro^®^ Adipogenesis, Osteogenesis and cholndrogenesis differentiation kits were applied according to the manufacturer’s guidelines (Gibco, Thermo Fisher Scientific, Waltham, MA, USA). After 21 days of upkeep, cells were fixed with 4% methanol-free formaldehyde (Molar Chemicals, Hungary) for 20 min at RT. The differentiation statuses of AD-MSCs were validated using different staining. For visualization of lipid-laden particles, Nile red staining (Sigma-Aldrich, Merck KGaA, Darmstadt, Germany) was applied, Alizarin red staining (Sigma-Aldrich, Merck KGaA, Darmstadt, Germany) was utilized to show mineral deposits during osteogenesis, and Toluidine blue staining (Sigma-Aldrich, Merck KGaA, Darmstadt, Germany) was used to label the chondrogenic mass.

### 2.3. Flow Cytometry

Characterization of the surface antigen expression pattern was implemented using three-color flow cytometry using fluorochrome-conjugated antibodies with isotype-matching controls. For fluorochrome signal measurement, the BD FACSAria TM Fusion II flow cytometer (BD Biosciences Immunocytometry Systems, Franklin Lakes, NJ, USA) was applied, and the data were analyzed using Flowing Software (Cell Imaging Core, Turku Centre for Biotechnology, Finland).

### 2.4. Treatment with LPS and TNFα Treatment for RNA Isolation

The AD-MSC cells utilized were derived from abdominal adipose tissue from three different donors. In a T25 cm^2^ flask, 2.8 × 10^5^ cells were seeded, using the cell culture medium described above, and the cells were maintained for 24 h. After that, the cell culture medium was replaced for treatment, and the cells were treated with (A) LPS [100 ng/mL] (ultrapure, Invivogen, San Diego, CA, USA) or (B) TNFα [100 ng/mL] (Peprotech, London, UK), and cells were incubated for 24 h under standard conditions (37 °C, 5% CO_2_, untreated cells left as control). Upon 24 h treatment, cells were collected and used for RNA isolation.

### 2.5. RNA Isolation for RNA Sequencing

Cells were collected, and the pellet was suspended in 1 mL TRI Reagent^®^ (Genbiotech Argentina, Bueno Aries, Argentina) and stored at −80 °C for 24 h. Upon thawing, 200 µL chloroform was measured in the samples, and after rigorous mixing, they were incubated at RT for 10 min. The samples were centrifuged at 13,400× *g* at 4 °C for 20 min for phase separation. The aqueous phase was transferred to clean tubes, and 500 µL 2-propanol was added and thoroughly mixed. The incubation and phase separation steps were then repeated. The supernatants were discarded, and the pellets were washed with 750 µL 75% EtOH-DEPC. The samples were centrifuged at 7500× *g* at 4 °C for 5 min, then the supernatants were removed, and the samples were dried at 45 °C for 20 min. The pellets were dissolved in RNase-free water and incubated at 55 °C for 10 min. The concentration was determined using an IMPLEN N50 UV/Vis nanophotometer (Implen GmbH, Munich, Germany) and the RNA samples were stored at −80 °C until use.

### 2.6. RNA-Sequencing

To obtain global transcriptome data, high-throughput mRNA sequencing analysis was performed on the Illumina sequencing platform. Total RNA sample quality was checked on Agilent BioAnalyzer using the eukaryotic Total RNA Nano Kit according to the manufacturer’s protocol. Samples with an RNA integrity number (RIN) value >7 were accepted for the library preparation process. RNA-Seq libraries were prepared from total RNA using the Ultra II RNA Sample Prep kit (New England BioLabs) according to the manufacturer’s protocol. Briefly, poly-A RNAs were captured by oligo-dT conjugated magnetic beads, and then the mRNAs were eluted and fragmented at a degree of 94 Celsius. First-strand cDNA was generated via random priming reverse transcription, and after the second-strand synthesis step, double-stranded cDNA was generated. After repairing ends, A-tailing and adapter ligation steps adapter ligated fragments were amplified in enrichment PCR, and finally, sequencing libraries were generated. The sequence runs were executed on the Illumina NextSeq 500 instrument using single-end 75-cycles sequencing.

### 2.7. Data Analysis

Gene expression analysis was performed in R (version 4.2.0). As a prefiltering step, we removed genes with low expression values (rows that only have 10 counts across all samples were removed) from further analysis. Principal component analysis (PCA) was used to visualize sample-to-sample distances. The PCA plot was created with the R package PCAtools, and it has not shown any significant batch effects. Differential expression analysis was performed using DESeq2 [36]. Significantly differentially expressed genes (DEG) were defined based on adjusted *p* values < 0.05 and log2-fold change threshold = 0.

The visualization of the DEG heat map was performed using the R package ComplexHeatmap [37], where Pearson correlation was used in rows and columns, and the z scores were calculated from normalized count data (normalization was performed using DESeq2 counts (dds, normalized = T). Volcano plots were created using the EnhancedVolcano package. In gene set enrichment (GSEA), DEGs were ordered by their log2-fold changes and used as input for gene set enrichment analysis. The R package ClusterProfiler was used with pvalueCutoff = 0.05 for both GO and KEGG GSEA. GO terms were further clustered using rrvgo with default settings. Heat maps of the different pathways were created based on preselected gene sets. DEGs that were part of each gene set were visualized with the same method as the heat map visualization of all the genes (for further details, see Appendix A). The DEGs were selected and ordered according to log2-fold changes, and this ordered list was used as input for the gene set enrichment analysis (GSEA). They were aligned with the Gene Ontology and KEGG databases.

### 2.8. Protein Array

Three AD-MSC donors were treated with LPS and TNFα, as described above. Supernatants were collected and stored at −80 °C until use. After thawing the samples, the supernatants were pooled by type of treatment and applied to the Human XL cytokine array Kit Proteome Profiler (R&D Systems, Biotechne, McKinley Place NE, Minneapolis, MN, USA) to determine secreted factors. The array was carried out according to the manufacturer. The images were quantified using Fiji (Image J 1.53S) with an embedded Protein Array Analyzer (Version:1.1.c) macro.

### 2.9. Wound Healing Assay

Three primary cell lines derived from abdominal adipose tissue were applied from three different donors. Cells were collected and counted using the EVE automatic cell counter (NanoEntek, Hwaseong, Republic of Korea). For the wound healing assay, 5 × 10^4^ cells were seeded per well of 24-well plates seeded in the upkeep medium. After seeding, cells were cultured for 24 h under standard conditions (37 °C, 5% CO_2_), and then the protocol was divided into two different lines. (I) After 24 h of incubation, the scratch was made, and the medium was immediately changed to a medium containing inflammatory agents, and it was followed by 48 h of time-lapse microscopy. (II.) After 24 h of incubation, the medium was changed to media containing inflammatory agents, and cells were incubated for 24 h under standard conditions, and then the scratch was made and the medium was changed immediately to agents-free maintaining media, and it was accompanied by 48 h of time-lapse microscopy. During treatment, six conditions were applied in both cases: (A) control (B) TNFα [100 ng/mL] and (C) LPS [100 ng/mL]. Scratches were generated using the AutoScratch Wound Making Tool (Agilent/BioTek, Santa Clara, CA, USA). Microscopic images were taken with Olympus Fluorescent Microscopy (Shinjuku, Tokyo, Japan) and analyzed by Fiji/ImageJ.

### 2.10. RNA Isolation for qPCR

Upon LPS and TNFα treatment (described above), the Macherey–Nagel NucleoSpin RNA Mini kit (Dueren, Germany) was applied according to the manufacturer’s instructions. Hereinafter, all work with RNA and cDNA samples was performed in a BioSan UVT-B-AR DNA/RNA UV-Cleaner box (Riga, Latvia).

### 2.11. Real-Time PCR

After extracting RNA and checking their quality and quantity with IMPLEN N50 UV/Vis nanophotometer, cDNA synthesis was carried out using High-Capacity cDNA Reverse Transcription Kit (Applied Biosystems^TM^, Thermo Fisher Scientific, Waltham, MA, USA) according to the manufacturer’s guidelines. For reverse transcription, Analytik Jena qTOWER^3^ G Touch Real-Time Thermal Cycler (Jena, Germany) was applied.

### 2.12. qPCR

The Xceed 2x Mix No-ROX kit (Institute of Applied Biotechnologies, Prague, Czech Republic) and TaqMan probes (250 rxns, FAM-MGB, Thermo Fisher Scientific, Waltham, MA, USA) were applied for quantitative PCR. Three technical and three biological replicates were used in all cases, and data were analyzed using the 2^−∆∆Ct^ method. The protocol was performed according to the manufacturer’s instructions, and the following probes were selected for the experiment: Hs00427620, Hs99999903, Hs00153133, Hs00361185, Hs 00194611, Hs00747615, Hs00174103, Hs00164932, Hs01001602, Hs 00171042, Hs00598625, Hs00265033, Hs00985639, Hs00165814, Hs01003372.

### 2.13. Cellular Impedance Measurement

Label-free cellular impedance measurements were performed with an Agilent xCELLingence Real-Time Cell Analysis (RTCA) DP (dual-purpose) instrument (Agilent/BioTek, Santa Clara, CA, USA). In total, 1 × 10^4^ AD-MSCs were seeded per well of E-Plate 16. Cells were incubated overnight under standard conditions (37 °C, 5% CO_2_) for attachment. The cells were then treated with LPS and TNFα (described above), and the cellular impedance was measured for 24 h (37 °C, 5% CO_2_).

## 3. Results

The isolated AD-MSC primary cell lines were characterized via trilineage differentiation and FACS analysis to validate their specific cell type properties (Appendix A).

To dissect the expressional changes and associated pathways that accompany the preconditioning of AD-MSC with LPS and TNFα preconditioning of AD-MSCs, we performed a differential gene expression analysis of the RNA sequence data. Furthermore, we employed distinct strategies in investigating signaling pathways: a hypothesis-driven gene set investigation of preselected terms and a hypothesis-free gene set enrichment analysis (GSEA) using GO and KEGG databases. RNA was collected after conditioning AD-MSCs with inflammatory factors for 24 h.

Both treatments had an effect on global gene expression (Appendix A); we identified 2752 DEG in TNFα and 1613 DEG in cells treated with LPS. Normalized expression values of these genes were clustered according to Pearson’s correlation in the control and treated samples and are shown on a heatmap (Figure 1A). To further dissect the differences between the two treatments, we investigated the number of genes overlapping up- and down-regulated (Figure 1B). There is a notable overlap in the effects of treatments, with 607 genes showing elevated expression and 449 genes showing decreased expression. The LPS treatment affected a similar number of genes up- and down-regulated, while TNFα slightly distorted the number of genes up-regulated (Figure 1C).

AD-MSCs have been shown to express cytokines, chemokines, and growth factors with wound-microenvironment-modulating capabilities, consequently promoting wound healing processes [38]. Interestingly, as an effect of preconditioning, we found several DEGs connected to these processes.

Among immunomodulatory factors released by AD-MSCs, IL1B, IL1RN, IL11, IL6, and IL15 showed elevated expression, while IL16 and TGFB3 showed decreased expression in a TNFα environment. LPS had less effect on immunomodulatory gene expression; it decreased the expression of IL1RN and TGFB3.

In addition, we observed changes in the expression of IDO and KYNU, which are associated with the apoptosis of T cells [2]. They were up-regulated in TNFα, while they were not significantly altered in LPS-treated stem cells. ICAM and VCAM play a role in lymphocyte recruitment to the injured site, both of which we found to be up-regulated in TNFα conditioning [2].

This shows that both treatments elicited immunomodulatory responses from AD-MSCs, potentially changing the behavior of these cells at the damaged tissue site to facilitate the healing process.

In addition to these molecules, chemokines that play a role in cell migration and immune regulation were also affected [39]. The expression of chemokines from members of the CXCL family (CXCL2, CXCL3, CXCL6, CXCL8, CXCL9, CXCL10, CXCL11) and the CCL chemokine ligand family (CCL1, CCL2, CCL5, CCL7, CCL20, CCL28) was strongly altered as a result of TNFα treatment, while LPS had very little effect on them.

In addition to immunomodulatory molecules, growth factors also play a crucial role in wound healing, especially in cell migration, proliferation, differentiation, and extracellular matrix synthesis [40]. Both treatments decreased VEGFB and TGFB3 and up-regulated VEGFC and FGF2. Additionally, TNFα up-regulated TGFA. These changes indicate that LPS and TNFα have an effect on MSC homing and tissue regeneration.

The repair of tissue damage by AD-MSCs requires the interplay of processes such as inflammation, proliferation, cell migration, and re-epithelization. These mechanisms often involve the same genes. Based on previously collected gene sets related to stem cell behavior in the wound healing process, we investigated the response of treatments to AD-MSCs and the overlap of these responses between gene sets.

We selected genes from the gene sets that were differentially expressed in at least one of the treatments; wherever we found identical genes, we made a connecting line between the sets represented by dots (Figure 2A). The widths of the lines represent the number of identical genes, while the size of each dot represents the fraction of DEGs in the entire gene set. In general, we can see that each set of genes is connected to at least five other sets and “wound healing migration”, “cellular senescence”, “cytokines growth factors”, “immune response”, and “stem cell EMT TEM” are connected to every other set, displaying dense interconnectedness.

Based on DEGs in most gene sets, LPS and TNFα treatment show comparable effects; consequently, as a result of hierarchical clustering, the two treatments form a distinct cluster from the control samples (Figure 2B). However, in three gene sets—“cellular senescence”, “immune response” and “cytokine growth factors”—clustering showed that LPS-treated AD-MSCs produced an expression pattern that is more similar to the control samples. Consequently, genes related to these terms showed greater expressional changes due to TNFα conditioning, while LPS had a much smaller effect.

To further investigate global gene expression changes, we conducted gene set enrichment analysis (GSEA) using two databases: Gene Ontology (GO) and Kyoto Encyclopedia of Genes and Genomes (KEGG). Similar results could be observed among the results of each database. The GSEA using GO terms resulted in redundant pathway terms due to the hierarchical structure of the database; therefore, we further clustered them based on semantic similarity (Appendix A). The parent terms within a group were chosen according to the adjusted p-values.

LPS and TNFα treatments had analogous effects in cell division, differentiation, cytoskeleton organization, and cell-cycle-related processes (Figure 3, Appendix A). Consequently, they gained an enhanced ability to increase cell proliferation, differentiation, and migration, which can be advantageous in wound healing.

Although the processes mentioned above change in the same direction, immune-related pathways seem to greatly differ in the two treatment-resulting changes. Among the GO clusters, LPS demonstrates regulation of tumor necrosis factor (TNF) production (Figure 3A). The first cluster contains: tumor necrosis factor production, tumor necrosis factor superfamily cytokine production, and relative regulation terms (Appendix A). GSEA resulted in negative normalized enrichment scores (NESs) for these pathways, therefore, indicating that the production of cytokines from the TNF and TNF superfamily is down-regulated. On the contrary, TNFα preconditioning affected several GO groups that are related to immune system processes: response to LPS, response to cytokines, regulation of cytokine production, receptor signaling through JAK-STAT, positive regulation of response to external stimulus, positive regulation of leukocyte cell–cell adhesion, positive regulation of leukocyte activation, negative regulation of cold-induced thermogenesis, humoral immune response, and granulocyte chemotaxis. All of these responses are up-regulated, showing an enhanced immune response.

These observations are also highlighted among the KEGG results (Figure 3B). In the case of TNFα, we can see signaling pathways, such as NOD-like receptor, NF-kappa B, IL-17, TNF, Toll-like receptor, Chemokine, and RIG-I-like receptor signaling pathways. These are all part of an immune response and have positive NESs. Furthermore, disease pathways such as influenza A, Hepatitis C, COVID-19, Measles, Legionellosis, Epstein–Barr virus infection and Herpes simplex virus 1 infection-including Toll-like receptor, RIG-I-like receptor, TNF, Jak-STAT, or NF-kappa B signaling pathways were also significant with positive NESs. In contrast, LPS treatment caused a different change and down-regulated COVID-19 and Shigellosis, while the immune pathways that were activated after TNFα treatment did not change significantly. Analysis with the QIAGEN IPA platform yielded similar results. For both treatments, the inflammatory environment manifested itself mainly in the regulation of the cell cycle, with effects on cell division and migration. The other biological functions were related to this, forming patterns that are most characteristic of tumor cells. We believe that this may be because the steps of tumor formation and metastasis are almost identical to those during wound healing, with similar stages observed (Appendix A).

The expression of certain genes was validated using quantitative real-time PCR, and the results show changes in the response of both treatments. As a result of the treatments, there were expression changes of genes that can be divided into different groups: STAT6, which is important in the initiation of signaling cascades; various cytokines: IL-6, CXCL-8, CXCL-10; factors involved in regeneration: TDO2, PTGS2, ICAM-1 and VCAM.1; and the expression of CCR4, which plays a role in migration, also changed. Both agents caused an elevation in the levels of CXCL8, IL-6, and PTGS2 proteins and a decay in STAT6 levels. Inflammatory factors altered the production of some proteins in the opposite direction, KRT14 (data not shown) was elevated and CCR4, TDO2, and VCAM-1 decreased with the use of LPS, while CXCL10 was increased upon TNFα treatment (Appendix A). The cellular impedance measurement showed that cell viability was not altered by the addition of inflammatory agents, and the seeded cells showed the same cell index before (12 h) and after (24 h, 36 h) treatment (Figure 4A). To underline the data for the analysis of gene expression, protein production was measured in AD-MSC supernatants after treatments with LPS and TNFα treatments. The protein array showed that Angiopoietin-2, CD40 ligand, Dkk1, FGF-19, HGF, ICAM-1, IGFBP-3, IL-17A, IL-22, IL-23, IL-24, LIF, MCP-3, MMP-9, RANTES, SDF-1α, Thrombospondin-1, uPAR, VEGF, Cystatin C, IL-6, IL-8, MIF, Osteopontin, and Pentraxin 3 protein expression increased with both treatments compared to the control. However, treatments have opposite effects on the production of a few proteins, such as LPS treatment decreasing DDPIV and GDF-15 expression, while TNFα decreases Apoliporotein A1, IL-4 and angiogenin synthesis (Figure 4B–D, Appendix A).

AD-MSCs are involved in all three parts of wound healing: the inflammation phase, proliferation phase, and remodeling phase. The wound healing assay demonstrated that MSCs promote faster wound closure after LPS and TNFα treatment compared to the control (Figure 5A). In that case, when treatment took place before woundmaking, the wounds of the control sample were closed around 40 h and the treated samples presented wound healing around 34.67 h after LPS and 37.67 h upon TNFα treatment. In the condition where scratching occurred before inflammation was triggered, the wounds of control samples closed around 47 h, and the treated samples closed faster, and they were remodeled around 41 h after LPS and 40.33 h after TNFα treatment. Graphs show significant changes in wound-closure speed between controls and in the presence of inflammation. It appeared that the closing speed was significantly higher in the case where inflammation induced after scratch formation was formed. AD-MSCs presented accelerated wound healing under inflammation conditions, and AD-MSCs were triggered by inflammatory factors to facilitate wound healing (Figure 5B).

## 4. Discussion

The regenerating and immunomodulatory ability of mesenchymal stem cells is already known, but the development of an effective therapeutic procedure is still pending. Applying animal models and in vitro cell cultures showed that MSC licensing can significantly improve tissue regeneration efficiency, reduce the degree of inflammation, and promote faster wound closure [6,9,13,30,31,35,41,42]. During our experiments, we treated human AD-MSC primary cell cultures with LPS and TNFα in order to induce an inflammatory microenvironment. Then, we examine the expression alterations at the gene and protein levels. We observed that both treatments resulted in significant changes in the RNA-Seq profile compared to the control. In the LPS and TNFα environment, the expression of genes related to cell proliferation, differentiation, wound healing, and migration increased. Furthermore, the TNFα treatment significantly elevated the expression of interleukins, chemokines, chemokine ligands and growth factors involved in the immune response. Furthermore, some interleukins showed an increased level in response to LPS treatment at the protein level. STRING analysis of secreted cytokines shows that LPS treatment induces the participation of the proteins MMP-9, VEGFA, ICAM-1, and ANGPT-2 in vascular remodelling, which is a major player in the overall wound healing process (Appendix A) [12,39,43,44]. HGF and FGF-19 proteins are involved in differentiation and, together with the LIF, CXCL-12, CCL7, and CCL5 chemokines, in the migration of MSC and local immunosuppression [12,39,43,44]. A similar pattern and pathways were activated by TNFα treatment, but here the activation of the immune response was more pronounced, with CD40L activation being more characteristic of pro-inflammatory MSC. Taken together, these data suggest that MSC activation can result in stronger wound healing. This confirms the hypothesis that licensing (preactivation) of cells may have therapeutic benefits. Pathways identified via gene expression studies were also confirmed using cytokine expression. Here, we have to take into account the limitation that much more information and pathways are available when examining gene expression data than when identifying 105 cytokines. Therefore, pathways based on gene expression data are more detailed and take into account the link to the cell cycle.

In the wound healing assay, both treatments significantly accelerated wound closure in the ‘treatment after scratch’ condition, and LPS also triggered faster wound closure in the ‘treatment before scratch’ condition, supporting our previous hypothesis. In general, both treatments were beneficial in terms of tissue regeneration, wound healing, and immunomodulation. It is promising, but more studies are needed to develop a possible therapeutic procedure. According to another approach, by developing our experimental setup, we can prepare a preclinical test. This may be suitable for tracking gene expression profile changes caused by the inflammatory microenvironment in the cells of patients with nonhealing wounds, ulcers, or burns. With our improved system, long-term personalized autogen therapy can be achieved.

Our results confirm that AD-MSCs can be used to treat chronic, inflamed, nonhealing wounds, even if the inflammation persists for a long time. While it is true that in the latter case, there is a difference in the processes induced by LPS and those in the presence of TNFα, from a clinical point of view, all treatments activate pathways that can promote tissue healing, ECM remodeling, and regeneration.

AD-MSCs have great clinical therapeutic potential due to their regenerative, antiapoptotic, antifibrotic, antioxidant, and immunomodulating capabilities. The delivery of not only AD-MSCs but also their secretome can have a positive effect on the course of the disease. Research also reveals that the secretion profile of AD-MSCs changes during appropriate pretreatments, making them even more suitable for therapy [21,42]. There is evidence in the literature that certain treatments can improve the effectiveness of MSCs. Most of these strategies seek to mimic the inflammatory medium. The effect of pro-inflammatory cytokines, hypoxia, has, in some cases, been shown to enhance MSC-mediated anti-inflammatory processes [45,46,47,48,49]. However, in these cases, classical pro-inflammatory signals were also detected during the induced response, making it unclear whether immunosuppressive or pro-inflammatory MSCs are formed and responsible for the possible therapeutic effect [45,50,51]. These indications have not yet been fully developed, as neither the type of agents nor their therapeutic concentrations are known or standardized. Suboptimally, a pro-inflammatory phenotype may be present more frequently, while high concentrations affect cell viability [50,52]. Pretreatments in GMP manufacturing of cell therapy products may also raise licensing issues. However, it should be noted that licensed MSCs are often considered the next generation of MSC therapies for the treatment of injuries associated with acute and sub-acute inflammation [50,52,53]. Research into the background of biological processes and safe treatments is still ongoing, but the results so far are extremely promising [21,42].

## Figures and Tables

**Figure 1 cells-12-01966-f001:**
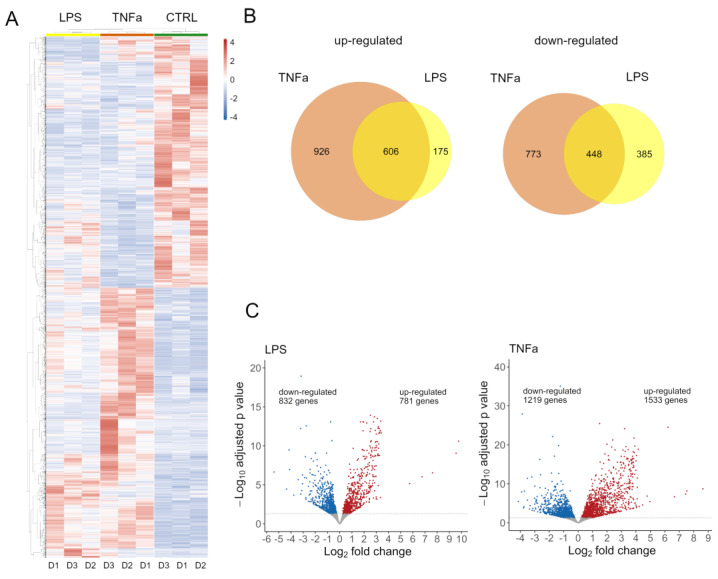
(**A**) The result of the RNA sequencing evaluation can be seen on the heat map. Gene expression changes of three human AD-MSC donors can be observed under treated conditions with control (green), LPS (yellow), and TNFα (red) treated conditions. The three conditions can be well separated, but there are some similarities between them. (**B**) Venn diagram (left) represents the number of genes with increased expression by TNFα (brown) and LPS (yellow), and the fitted cross-section shows genes that were affected by the two treatments. The Venn diagram (right) represent the number of genes with reduced expression due to TNFα (brown) and LPS (yellow) treatment; the adjusted cross-section shows genes affected by the two treatments. (**C**) Volcano plots show the genes with significantly increased or decreased gene expression that changed the most as a result of treatments. Genes with the highest reduced expression (blue) and those with increased expression (red) are shown.

**Figure 2 cells-12-01966-f002:**
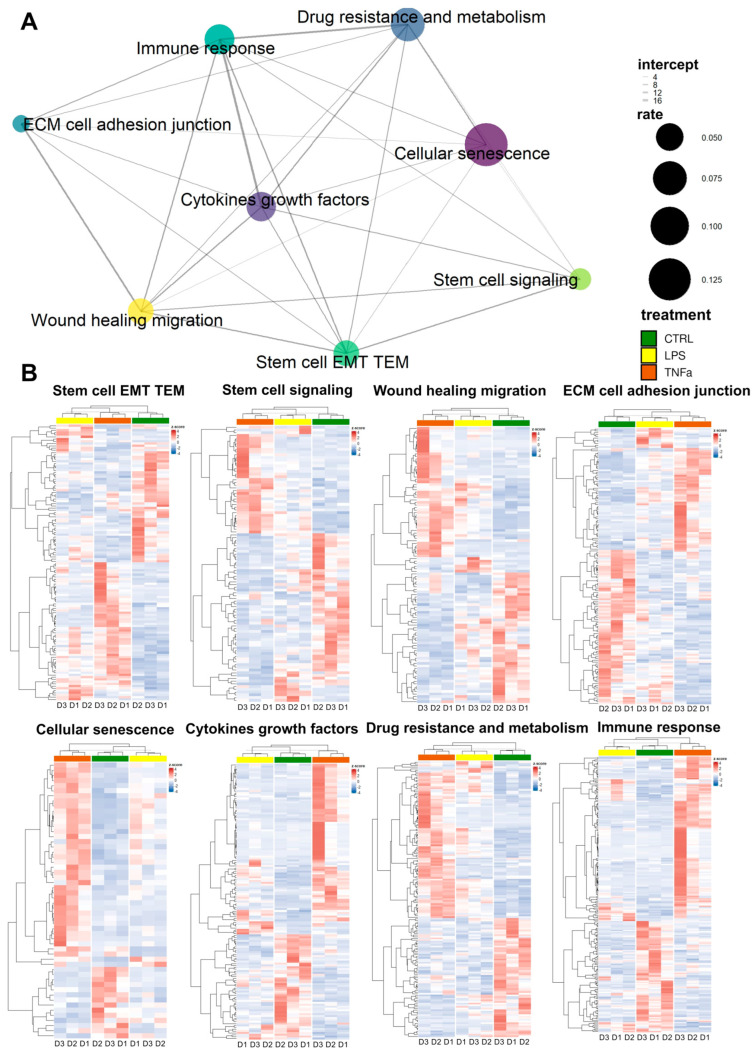
(**A**) The enrichment analysis of the gene set shows the percentage of genes in the wound healing-related pathway that changed as a result of the treatments. The size of the circles is directly proportional to the proportion of changed genes, and the thickness of the lines between them is directly proportional to the overlap of the genes. (**B**) The changes in gene expression of the pathways shown in Figure 2A can be seen in detail in the heat maps. Control (green), LPS (yellow), TNFα (red).

**Figure 3 cells-12-01966-f003:**
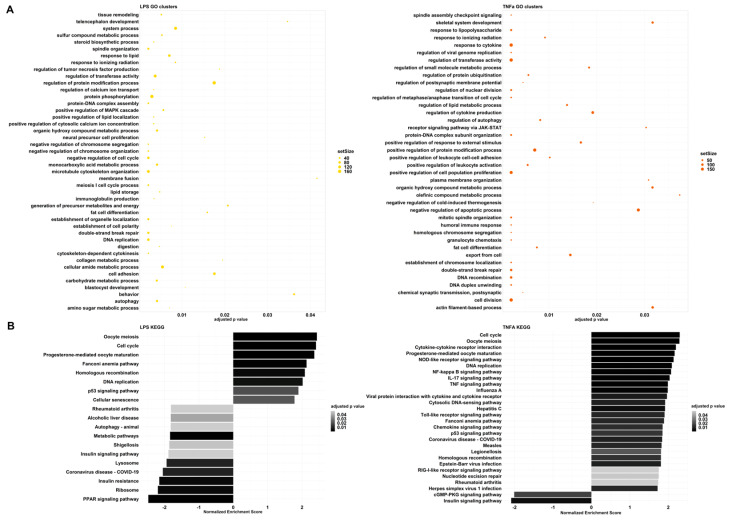
(**A**) Parent terms of the results of the gene set enrichment analysis based on GO terms in the LPS and TNFα treatments. The x-axis represents adjusted *p* values; the size of the dots shows the size of each GO term’s gene set. (**B**) Results of the gene set enrichment analysis based on KEGG in LPS and TNFα treatments. The pathways are ordered according to their normalized enrichment score, and the color intensity of the columns depends on the adjusted *p*-value.

**Figure 4 cells-12-01966-f004:**
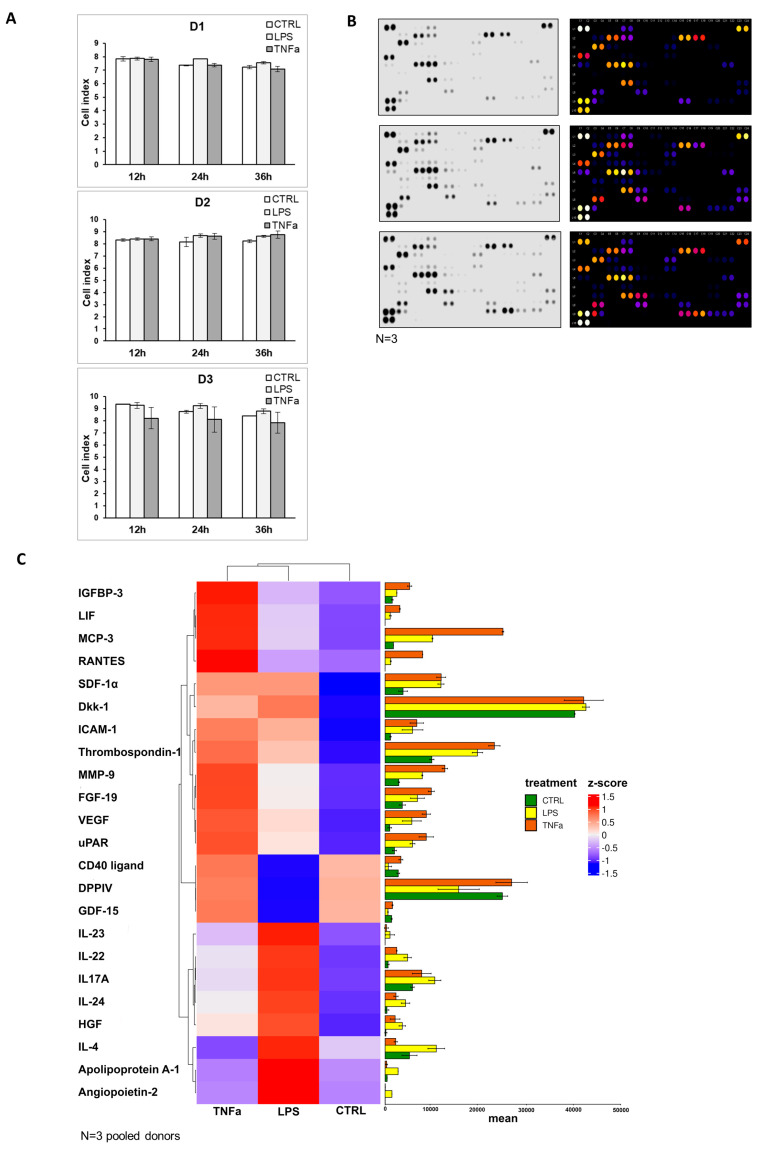
(**A**) As a result of impedance measurement, it can be seen that the cell index did not change significantly during treatment (24 h, 36 h) compared to the untreated state of the cells (12 h). From this result, it can be concluded that the viability of the cells remained unchanged during the treatment. Control (white), LPS treatment (light gray), and TNFα treatment (dark gray). (**B**) The result of the protein array is the original image of the membrane, followed by the Fiji diagram created to determine the pixel density. (**C**) The visual representation of the protein level changes from the protein array results can be seen on the heat map, as well as on the protein bar chart showing the greatest changes after treatment, control (green), LPS treated (yellow), and TNFα-treated (red). (**D**) Detailed representation of the protein array results with the attachment of the original membrane photo. Bar graphs show individual changes per protein, control (white), LPS-treated (light gray), TNFα-treated (dark gray).

**Figure 5 cells-12-01966-f005:**
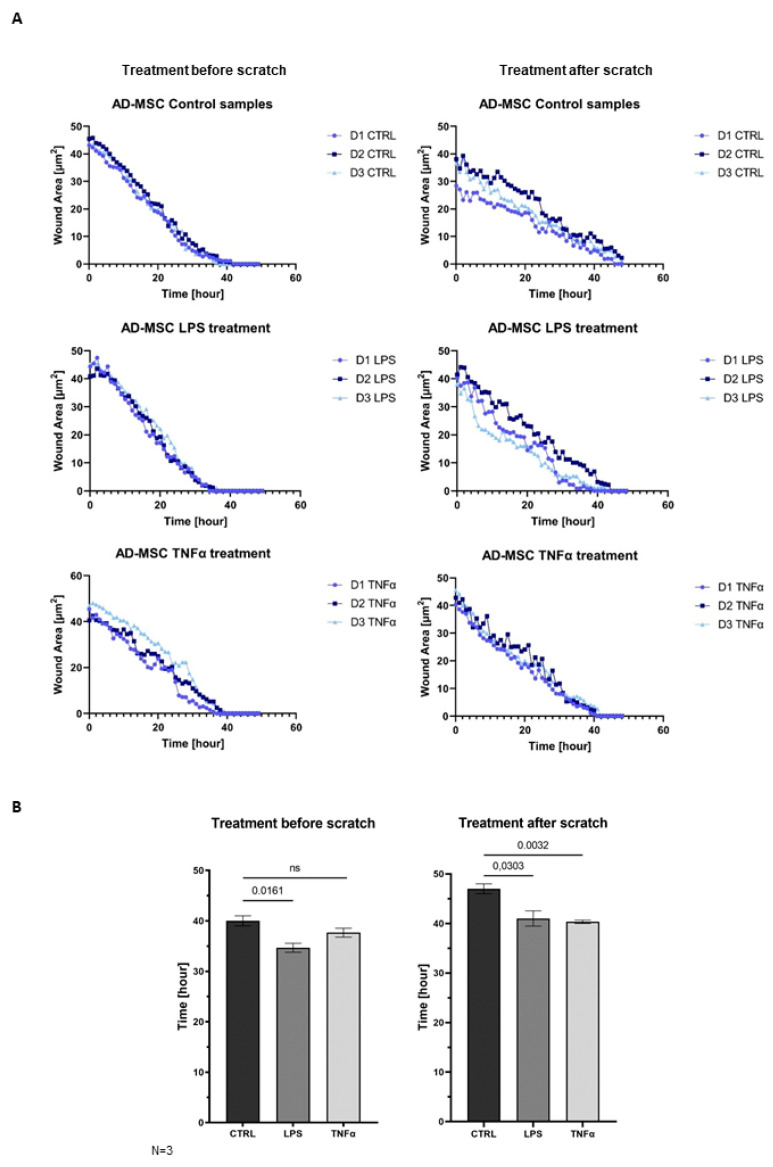
(**A**) The result of the wound healing test shows the area of the wound as a function of time. The wound closure process was monitored for 48 hours. Each donor is marked with a different color: D1 (medium blue), D2 (dark blue), and D3 (light blue). (**B**) Bar graphs show comparison of wound closure length between conditions - treatment before and after scratching.

## Data Availability

All data generated and analyzed during this study are included in this manuscript (and its Appendix A files).

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
