# Peer review of "Effect of Inflammatory Microenvironment on the Regenerative Capacity of Adipose-Derived Mesenchymal Stem Cells"

_cells, 2023, doi:10.3390/cells12151966_

Round 1

Reviewer 1 Report

Review of 'Effect of inflammatory microenvironment on the regenerative capacity of adipose-derived mesenchymal stem cells'

Szűcs et al. investigate the effect of an inflammatory microenvironment on the regenerative capacity of adipose-derived mesenchymal stem cells (AD-MSCs). The authors show high-throughput gene expression assays on AD-MSCs activated with LPS and TNFα and analyze the RNASeq data to identify distinct gene expression patterns and biological pathways. The authors further examine secreted cytokines important in the immunological response at the protein level and perform a functional assay to assess wound healing. Together, the manuscript shows that AD-MSCs presented accelerated wound healing under inflammation conditions, suggesting the use of these culture conditions in clinical applications.

Overall, the authors performed several assays and produced a variety of data, and their findings are overall interesting. However, there is a lack of novelty as similar results were previously published (Some examples: Fu, X., Han, B., Cai, S., Lei, Y., Sun, T., and Sheng, Z. (2009). Migration of bone marrow-derived mesenchymal stem cells induced by tumor necrosis factor-α and its possible role in wound healing. Wound Repair and Regeneration 17, 185191. https://doi.org/10.1111/j.1524-475x.2009.00454.x. Munir, S., Basu, A., Maity, P., Krug, L., Haas, P., Jiang, D., Strauss, G., Wlaschek, M., Geiger, H., Singh, K., et al. (2020). TLR4dependent shaping of the wound site by MSCs accelerates wound healing. EMBO reports 21. https://doi.org/10.15252/embr.201948777.)

Additionally, several points should be addressed

Main points:

1)                  The introduction lacks a lot of details about the current state of the field, and the hypothesis and goals of the study are not well explained. Additionally, the references are not allocated properly and accurately and ignore most of the relevant literature in the field that deals with the term 'licensing'.

2)                  An in-depth discussion regarding the differences in the transcriptional changes and the cytokine set between the TNFa and the LPS treatment could add value and novelty to the manuscript.

3)                  Please add in all figures statistics including significance p value and number groups (n). It is not possible to evaluate the verity of some of the author’s statements.

4)                  Line 396 “They were remodeled around 20-32h”: There is no evidence of this in the graphs, a better explanation is required.

5)                  The discussion does not contain a critical scientific discussion on the authors’ results or the importance of their findings. It is just a review of the literature.

Minor edits:

1)                  Lines 300-345 are filled with irrelevant information- please reduce and summarize results and concentrate on the main pathways.

2)                  Fig. 6 is repetitive put in supplementary or move fig. 5

3)                  Some of the genes discussed in line 288-289 are not present on the volcano graphs.

in general, the authors have many grammar mistakes

4)                  In general authors should be aware of their inappropriate use of could instead of can.

5)                  30-31 should be “We also mapped the biological pathways by further investigating the most altered genes, using the Gene Ontology and KEGG databases”

6)                  55 should be “ability to differentiate into”

7)                  66 should be “showed in the presence”

8)                  89 should be “ensure safe and”

9)                  267 should be “distinct conditions. Based on these results, the data set can”

10)               269 should be “data set. Only”

11)               340 should be “TNFα positively affects”

12)               360 should be “Donors apparently behave differently in response to the treatments, but the direction and extent of the effect is the same” sentence is contradictory

Reviewer 2 Report

Szűcs et al investigate the effect of a pro-inflammatory environment on adipose-derived mesenchymal stem cells (AD-MSC). Using RNA Seqeuncing, protein array and wound healing assays the evaluate the regenerative capacity of AD-MSC differentiated from 3 different donors stimulated with LPS and TNFa. Their results show that stimulation with LPS and TNFa alters gene expression profile and accelerates wound healing.

While the methods are described detailed all other section should be improved. While the introduction gives background on the topic it misses a clear statement about the aim of the study.

With regards to the presentation and description of the results, I have some mayor points to raise:

All results (including stainings from differentiated cells to prove their identity (Supplementary Figure 2A/2B) should be shown and reported in the results section of the manuscript but not in the method section.

In order to understand the rational of the performed experiments, the results section should introduce the experiments and the treatments rather than plainly explaining the obtained results with no context/further information

The figure legends should provide additional information  to understand the figures rather than in interpretation of the results such as in line 812: As a result of the impedance measurement, it can be seen that the viability of the cells remained unchanged during the treatment.

Supplementary figures are not available

Some figures are barely readable

Regarding figure 4: While the authors claim, that the “results show changes in response of both treatments” in the figures themselves, no significance was stated. Can the authors clarify, if there are no significant changes, or if these are simply no changes .

So here I wonder if the the conclusion are indeed supported by the results.

The discussion section reads like a summary of effects of different cytokines on wound healing rather than fulfilling its function as a discussion: Putting the observed results into a broader context

Minor:

There are several English language issues

Line 166-180: RNA sequencing: line spaing is too big

Line 215: three cell lines instead of cell line

there are some issues to be fixed (mainly in the intro)

Round 2
